# Human cell based directed evolution of adenine base editors with improved efficiency

Junhao Fu[1,5], Qing Li [2,5], Xiaoyu Liu[1,5], Tianxiang Tu[1,5], Xiujuan Lv[1], Xidi Yin[2], Jineng Lv[1], Zongming Song [1,3], Jia Qu[1], Jinwei Zhang [4], Jinsong Li [2✉] & Feng Gu [1✉]

Adenine base editors (ABE) are genome-editing tools that have been harnessed to introduce precise A•T to G•C conversion. However, the low activity of ABE at certain sites remains a major bottleneck that precludes efficacious applications. Here, to address it, we develop a directional screening system in human cells to evolve the deaminase component of the ABE, and identify three high-activity NG-ABEmax variants: NG-ABEmax-SGK (R101S/D139G/E140K), NG-ABEmax-R (Q154R) and NG-ABEmax-K (N127K). With further engineering, we create a consolidated variant [NG-ABEmax-KR (N127K/Q154R)] which exhibit superior editing activity both in human cells and in mouse disease models, compared to the original NG-ABEmax. We also find that NG-ABEmax-KR efficiently introduce natural mutations in gamma globin gene promoters with more than four-fold increase in editing activity. This work provides a broadly applicable, rapidly deployable platform to directionally screen and evolve user-specified traits in base editors that extend beyond augmented editing activity.

[1] School of Ophthalmology and Optometry, Eye Hospital, Wenzhou Medical University, State Key Laboratory and Key Laboratory of Vision Science, Ministry of Health and Zhejiang Provincial Key Laboratory of Ophthalmology and Optometry, Wenzhou, Zhejiang, China. [2] State Key Laboratory of Cell Biology, Shanghai Key Laboratory of Molecular Andrology, Shanghai Institute of Biochemistry and Cell Biology, Center for Excellence in Molecular Cell Science, Chinese Academy of Sciences, Shanghai, China. [3] Henan Eye Hospital, Henan Eye Institute, Henan Provincial People's Hospital and People's Hospital of Zhengzhou University and People's Hospital of Henan University, Zhengzhou, Henan, China. [4] Laboratory of Molecular Biology, National Institute of Diabetes and Digestive and Kidney Diseases, National Institutes of Health, Bethesda, Maryland, USA. [5]These authors contributed equally: Junhao Fu, Qing Li, Xiaoyu Liu, Tianxiang Tu. ✉email: jsli@sibcb.ac.cn; gufenguw@gmail.com

Recent advances in genome engineering technologies based on CRISPR-Cas9 are enabling the systematic interrogation of mammalian genome functions and numerous targeted medical applications[1–3]. Cas9-triggered double-strand DNA break (DSB) at a target sequence in the genome is subsequently repaired by nonhomologous end joining (NHEJ), microhomology-mediated end joining (MMEJ), or homology-directed repair (HDR)[1,4]. CRISPR-Cas9 incurs toxicity because it generates DSBs and off-target mutations that are more prone to occur when a nuclease-induced DSB is present[5]. Base editing is a genome-editing method that directly generates precise point mutations without creating DSBs[6]. As base editors do not produce DSBs, they minimize the off-target effect, compared with CRISPR-Cas9 mediated editing[2].

The current base editing technology can be classified into DNA and RNA base editors. As to the first one, two classes of DNA base editor have been reported: cytosine base editors (CBE) which convert a C•G base pair into a T•A pair, and adenine base editors (ABE) which convert an A•T base pair into a G•C pair[2,3]. CBE and ABE can mediate all four possible transition mutations (C to T, A to G, T to C, and G to A). Base editors have been successfully implemented in various organisms including human cells, animals, and plants[6–12]. Nevertheless, the gene-editing utility of base editors is hampered by low efficiency and the off-target effect[13,14]. ABE are particularly limited due to its lower efficiency compared with CBE[15].

To overcome these limitations, several strategies have been adopted to increase the activity of ABE. These include optimizing the codon usage of ABE, modifying the location and number of nuclear localization signals (NLS), reconstructing ancestral deaminase component, and using different forms of sgRNA[10,16–19]. These extensive efforts reflect an urgent need to optimize ABE as an effective gene-editing tool. Additional questions also need to be addressed, including the control of the editing window for improved target site selection and reduction in RNA editing levels. By installing E59A mutation in the wild-type TadA domain (TadA) and V106W mutation in the evolved TadA domain (TadA*), or F148A mutation in both TadA and TadA*, engineered ABEs possess the higher fidelity, as to the RNA editing side effect[20,21]. The latter also has narrowed editing window[21]. However, for certain sites, such as when the target base exceeds the canonical editing window relative to the available PAM (protospacer adjacent motif) or the gene is inactivated by introducing premature termination codons, a wider editing window may be more desired. Here, with directed evolution in human cells, we sought to establish a general screening platform to engineer NG-ABEmax to identify variants with special traits, i.e., enhanced activity.

## Results

**Generation of a high-throughput *EGFP*-based screening system for adenine base editors**. To engineer and augment the existing adenine base editors, we first sought to generate a human cell-based reporter system that allows for rapid and accurate determination of ABE activity and specificity. We previously established an EGFP-reporter system in human cells to evaluate the efficiency of CRISPR-Cas9 and CRISPR-Cas12a[22–28]. Herein, when the sequence-specific RNA-guided endonuclease is targeted to the coding region of *EGFP*, it may generate frameshift indel mutations that lead to loss of fluorescence. Recently, GFP-based reporter systems have been developed for assess base editing[10,29,30]. However, such screening systems are ineffectual to directly evaluate ABE mutants. The reason for that is the editing of one or more adenosines to guanosines at different regions of the *EGFP* sequence leads to diverse local protein alterations that can exhibit highly variable effects on *EGFP* folding, structure, and consequently its fluorescence, thus confounding analysis by cytometry.

To create a screening system where base editing completely ablates, rather than unpredictably alters *EGFP* activity, we first tested if ABE editing of a tryptophan TGG codon in *EGFP* (W58) to an amber stop codon TAG inactivates *EGFP* by truncating the protein near its N-terminus[10] (Supplementary Fig. 1a). We observed that the green fluorescence of *EGFP* was abolished by this conversion, suggesting the feasibility of this approach (Supplementary Fig. 1b). To avoid any potentially confounding effects of editing neighboring nontarget adenosines that flank the W58 site, we then sought to replace all adjacent adenosines with other nucleobases while maintaining *EGFP* activity (Fig. 1a). While the 13 nucleotides upstream of W58 contained no adenosines, the 18 nucleotides downstream of W58 harbor three adenosines, all part of ACC codons encoding Threonine. By mutating the three ACC codons to TCG, we introduced three conservative substitutions: T60S, T63S, and T64S, and found that the mutations do not appreciably impact *EGFP* fluorescence (Fig. 1a–c). Lastly, we created a 5′-TG-3′ PAM (protospacer adjacent motif) site necessary for Cas9 recognition through a silent mutation (CTC-to-CTG coding for L61). We term this engineered *EGFP* target gene that harbors all four mutations "*EGFP* variant," which exhibited comparable EGFP expression levels as the wild-type (Fig. 1a–c, Supplementary Fig. 2).

Base editors such as ABE exhibit variable editing efficiencies at different adenosine sites within the editing window[3]. To measure the positional effects on editing efficiency and to identify an optimal site for subsequent variant screening, we generated 12 partially overlapping single guide RNAs (sgRNAs) that allow the single editing site to shift gradually from the 5′ end (position A1), to near the 3′ end (position A16) of a 20-nucleotide sliding window. To enable single-nucleotide step scanning in a compact editing window (Fig. 1d, e), we used NG-ABEmax to edit either an amber stop codon (TAG, *dEGFP1*) or an opal stop codon (TGA, *dEGFP2*) back to TGG (encoding W58), thus restoring full-length *EGFP* transcription and leading to an increase in fluorescence (Fig. 1f, g).

Consistent with the previous reports[3], our edit site scan analysis revealed a wide editing efficiency distribution centered around A6 and A7 with efficiencies ranging from 0.05% (A16) to 48% (A6), suggesting both robust editing efficacy and high site selectivity of the assay system (Fig. 1f, g).

**Evolution of NG-ABEmax variants with higher editing activity**. To demonstrate the practical utility of our stop codon reversion-based screening system for adenine base editors, we used it to screen for NG-ABEmax variants that exhibit augmented editing efficiencies. To this end, it is appropriate to select an editing site with low-to-intermediate editing efficiency with the current NG-ABEmax, so that any significant activity enhancement can be captured and accurately quantified. We selected A12 site with a 3.1% efficiency (Fig. 1f), which also falls outside the canonical editing window of NG-ABEmax, to screen for variants with either a generally augmented editing activity, or an ability to edit in a shifted window.

We first generated two random variant libraries of NG-ABEmax by error-prone PCR and named them library 1 and library 2 (Fig. 2a, Supplementary Fig. 3a), which carry mutations in the evolved TadA domain (TadA*) or the wild-type TadA domain (TadA), respectively. Library 1 produced about 1428 colonies (numbered L1-1 to L1-1428), and library 2 had 550 colonies (numbered L2-1 to L2-550). To estimate the frequency of mutagenesis, we randomly picked 20 colonies per library for Sanger sequencing. Sequencing analysis revealed that 19 of the

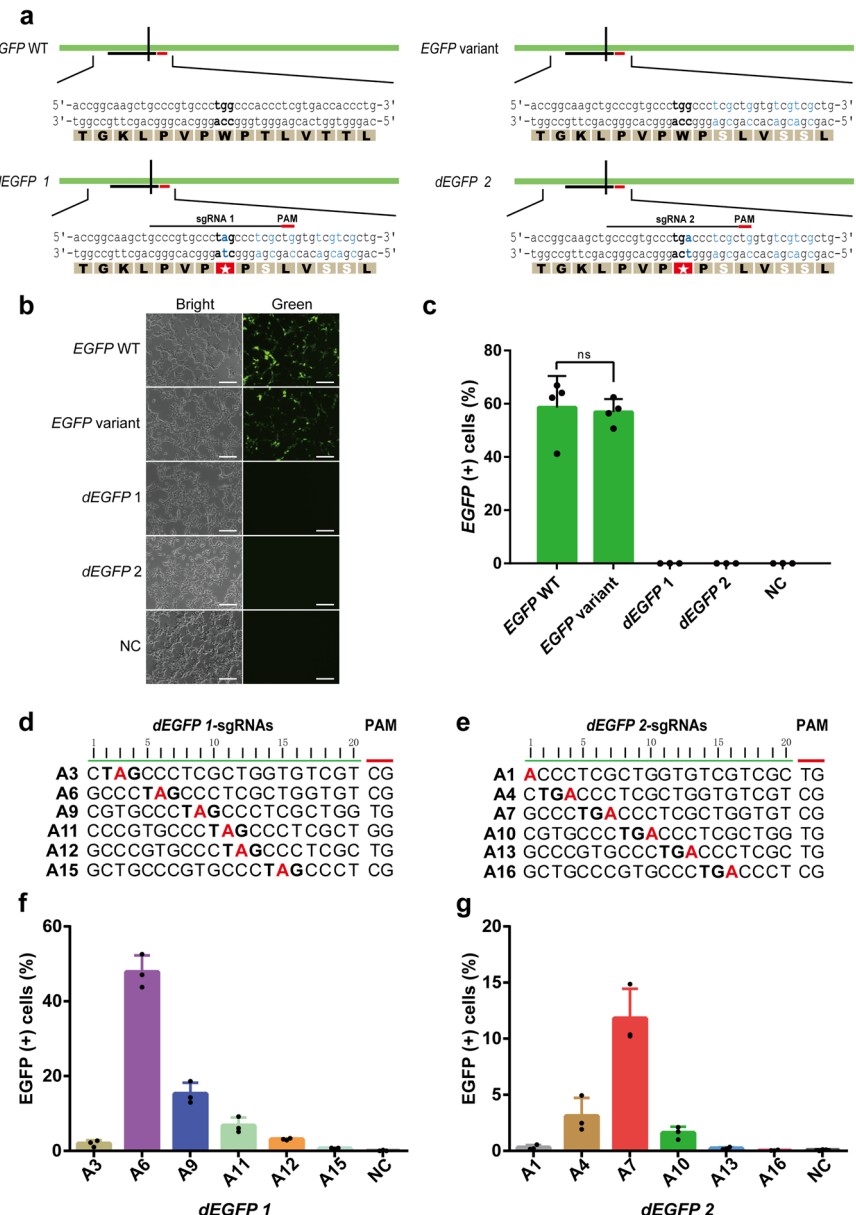

**Fig. 1 Design of the stop codon reversion-based screening system. a** Different *EGFP* variants. Single-nucleotide conversions are indicated by blue and residue substitutions are indicated by white. The stop codon was highlighted with red pentagram. **b**, **c** Fluorescence image (**b**) and editing efficiencies (**c**) of HEK-293 cells co-transfect with *EGFP* WT, variant, *dEGFP* 1 and *dEGFP* 2, respectively. *EGFP* variant maintains the expression, while *dEGFP* 1 and *dEGFP* 2 abolish the expression. (**c**) EGFP positive cells were quantified by flow cytometry. Scale bar, 10 μm. Data are shown as the average mean ± s.d. (*n* = 4 independent experiments). **d**, **e** The sequence of different sgRNAs (5-'NG-3' PAM) for *dEGFP* 1 and *dEGFP* 2, respectively. The stop codon and target A base have been highlighted in bold and red. **f**, **g** Editing efficiencies of A to G mutations at different positions of *dEGFP* 1 and *dEGFP* 2. Data are shown as the average mean ± s.d. (*n* = 3 independent experiments). Source data are available in the Source data file.

20 colonies from library 1 harbored unique mutation(s), while the 20th colony had a single-nucleotide deletion. Very similar results (19 colonies with unique mutation(s); one with deletion) were obtained with library 2. Further analysis revealed an average rate of ~4.8 substitutions per kilobase in these two libraries (Supplementary Table 1).

Subsequently, each variant (L1-1 to L1-1428 and L2-1 to L2-550 plasmids) was individually co-transfected with the corresponding sgRNA (*dEGFP1*-A12) and *dEGFP1* expression cassette into HEK-293 cells (Supplementary Fig. 3b), respectively. We found that 12 (8 of library 1 and 4 of library 2) out of the 1978 tested variants exhibited significantly elevated activity at A12, showing at least a threefold increase in editing activity

(8.9–11.25%), compared to the wild-type NG-ABEmax (2.62%-3.4%) (Fig. 2b, Supplementary Table 13, 14). These findings suggest that the screening platform is effective at identifying and enriching high-activity base editor variants.

To distinguish whether these 12 high-activity variants possess generally higher activity at all target sites or shift or expand the editing window to gain activity at the A12 site, we took advantage of our recently developed method for assessing base editing in human cells, in which editing efficiency is measured by agarose gel analysis of editing-mediated inactivation of restriction enzyme sites[15]. We generated a sgRNA that can simultaneously target a PstI restriction site (A7) and a SalI restriction site (A12) in the MCS (multiple cloning sites) sequence, which was integrated into the genome of

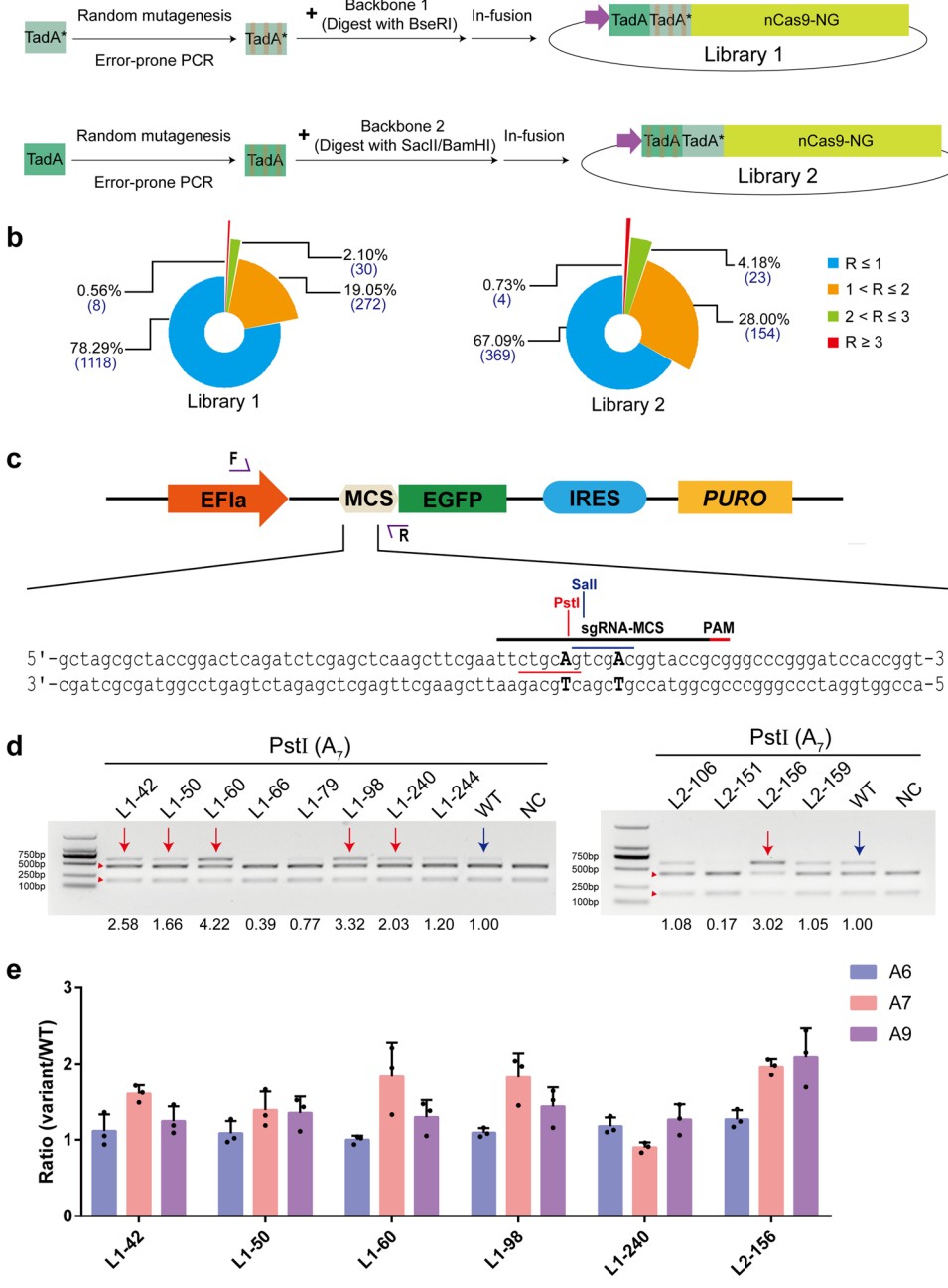

**Fig. 2 Evolution of NG-ABEmax variants with improved A•T to G•C base editing activity. a** Scheme of libraries generation design. The library 1 contain TadA* mutations and the library 2 contain TadA mutations. Red lines represent the point mutations. **b** Summary of the library 1 and library 2 (A•T to G•C base editing efficiencies at A12). R represents the ratio of editing efficiency of mutants to NG-ABEmax. **c** Schematic for assessing A•T to G•C base editing efficiencies of NG-ABEmax variants using the HEK293-PME cell line. Restriction sites are highlighted in red and blue. **d** Agarose gel electrophoresis results of testing editing efficiency of NG-ABEmax variants with PstI. Cleaved bands from PstI are labeled with red triangle. The amplicon is 606-bp. The variants with high-activities are highlighted with red arrows. WT represents NG-ABEmax and highlighted with blue arrows. **e** Boosted editing efficiencies of selected variants via *EGFP*-based reporter system. Error bars indicate mean ± s.d. ($n = 3$ independent experiments). Source data are available in the Source data file.

HEK-293 (termed HEK293-PME) (Fig. 2c). Each of these 12 variants was then co-transfected with this sgRNA into HEK293-PME cells. At 48 h post-transfection, cells were collected and isolated genomic DNA. PCR was performed with the genomic DNA as a template with specific primers. Purified PCR products were treated with PstI (178 + 428 bp) or SalI (177 + 429 bp), respectively, and then analyzed by agarose gel electrophoresis to estimate editing efficiency. Remarkably, L1-60, L1-98, and L2-156 showed significantly improved activity (more than ~threefold increase) at A7 site (Fig. 2d). While it may be due to the low

sensitivity of this restriction enzyme digestion-based method, we can't observe the boosting effects at A12 site, (Supplementary Fig. 4). The enhanced editing efficiencies of L1-60, L1-98, and L2-156 were further confirmed with above-described *EGFP*-based flow cytometry system (Fig. 2e). Notably, L2-156 possesses the highest activity among all tested variants, with ~twofold increase in editing efficiencies at A7 and A9 sites. These results suggest that these three variants possessed across-the-board higher editing activities both within and outside of the canonical editing window, rather than merely shifting the editing window.

**Mapping the key residues underlying enhanced editing activity.** To map the exact sequence changes in L1-60, L1-98, and L2-156, we sequenced the TadA* domains of L1-60 and L1-98 and the TadA domain of L2-156. We found multiple amino acid substitutions in L1-60 (R101S/D139G/E140K), L1-98 (E9K/Q154R), and L2-156 (A106S/N127K/E155K), respectively. To deconvolute any combinatory effects and identify key residues that drove higher editing activities in L1-60, L1-98, and L2-156, we dissected all possible single, double, and triple combinations of mutations found in these three variants using our *EGFP*-based system (Fig. 3a and Supplementary Fig. 5). We also sequenced the TadA* domains of the remaining eight variants and summarized the mutations information (Supplementary Table 2).

For L1-60, the highest activity occurred when R101S, D139G, and E140K were present simultaneously (hereafter termed NG-ABEmax-SGK, Fig. 3a). For L1-98, the performance of Q154R alone was comparable to that of E9K/Q154R. Therefore, Q154R is the key mutation for the enhanced activity of L1-98, and the variant harboring singe Q154R mutation termed NG-ABEmax-R (Fig. 3a). For L2-156, N127K alone accounted for essentially all the enhanced activity. The activity enhancement was completely abolished when N127K was reverted back to WT. Thus, N127K substitution was the driving force for the enhanced activity in L2-156 and is hereafter referred to as NG-ABEmax-K (Fig. 3a). Meanwhile, NG-ABEmax-R and NG-ABEmax-K have slightly higher activity, compared with that of NG-ABEmax-SGK (Fig. 3a).

As these high-efficiency variants were identified and validated on an exogenous *EGFP* gene, we next verified that they also function similarly at endogenous genomic loci. To this end, we transfected HEK-293 cells with these two variants NG-ABEmax-R or NG-ABEmax-K to edit a panel of 19 genomic sites of different sequence context and composition. Upon editing, genomic DNA was extracted from harvested cells at 48 h post-transfection and next-generation sequencing was performed.

Compared to NG-ABEmax, NG-ABEmax-SGK, NG-ABEmax-R, and NG-ABEmax-K exhibited significantly improved editing efficiency at most genomic sites tested (Supplementary Fig. 6, 7). The enhancement was particularly pronounced at low-efficiency sites. For NG-ABEmax-R, the enhanced activity was most obvious at A4 (average: 14.62% of NG-ABEmax-R versus 2.94% of NG-ABEmax), an increase of 5.0-fold (Supplementary Fig. 7b). For NG-ABEmax-K, the average activity increased by 2.2-fold at A7 where the enhanced activity was most evident compared to NG-ABEmax (average: 28.96% of NG-ABEmax-K versus 13.04% of NG-ABEmax) (Fig. 3b, Supplementary Fig. 7c). Mirroring the findings on the *EGFP* gene, the editing activity of these variants on genomic sites had been significantly improved both within and outside of the canonical editing window (Fig. 3, Supplementary Fig. 7). Notably, at the noncanonical editing window (i.e., A12), the activity of these variants remains very low, despite significant enhancement by the mutations (Fig. 3a). We also noticed that the editing activity of NG-ABEmax-K was highest among these variants at the majority of sites we examined (Fig. 3a. Supplementary Fig. 7). Taken together, we identified the key editing-enhancing residues of L1-60 (R101S/D139G/E140K), L1-98 (Q154R), and L2-156 (N127K), and showed that these substitutions led to general enhancement of editing activity at both exogenous and endogenous sites and both within and outside the canonical editing window.

**Combinatorial optimization of high-efficiency NG-ABEmax variants.** NG-ABEmax-K was selected for further study due to its highest editing activity. We hypothesized that replacing Asn127 in TadA with two other basic amino acids [Histidine (H) or Arginine (R)] may produce a similar effect due to a similar charge. We generated NG-ABEmaxH[127] and NG-ABEmaxR[127] and tested the performance (Supplementary Fig. 6). Interestingly, we found that while N127H had little effect on activity, N127R exhibited enhanced activity comparable to N127K within the canonical editing window (Supplementary Fig. 8a). This was most obvious at A7 (ranging from 2.73% to 64.98%, compared to NG-ABEmax [ranging from 1.3 to 48.1%)], with up to 6.02-fold increase on site 3 (Supplementary Fig. 8b). These data suggest that a long-chain basic residue at position 127 in TadA generally boosts the editing efficiency of NG-ABEmax, potentially by enhancing local interactions with the DNA substrate.

We next sought to further enhance the activity by combining mutations from NG-ABEmax-R, and NG-ABEmax-K, because using our EGFP-based system these two have slightly higher activity, compared with that of NG-ABEmax-SGK (L1-60, Fig. 3a). We generated a variant namely NG-ABEmax-KR (N127K mutation in TadA domain, Q154R mutation in TadA* domain) (Supplementary Fig. 6). We evaluated the activity at the endogenous genomic sites and found that NG-ABEmax-KR exhibited further increased activity (Fig. 3b, Supplementary Fig. 9). Specifically, the average activity of NG-ABEmax-KR was increased by 1.1–5.4-fold within the canonical editing window, with an activity ranging from 40.87 to 64.22% (7.6% to 60.27% for NG-ABEmax) (Supplementary Fig. 9). Upon additional examination, we found that the editing window has been greatly expanded in both directions in NG-ABEmax-KR (Fig. 3c–e, Supplementary Fig. 10). Specifically, compared with canonical editing window at the A4~A7, the NG-ABEmax-KR has an editing window at A3~A7, with a substantially higher activity at A3, A4, A6, and A7. These findings show that the original mutations acted partially or wholly independently of each other, thus allowing further enhancements by combinatorial optimization.

**NG-ABEmax-KR exhibits superior efficacy in mice disease models and human gene therapy.** To test whether NG-ABEmax-KR could be harnessed for efficient generation of mouse disease models, we designed an sgRNA to target c.202 of *Tyr* gene encoding the melanin-producing Tyrosinase, to introduce a missense codon (CAT to CGT, H420R) and generate an albinism mouse model[31] (Fig. 4a). Successful editing would produce *Tyr* mutant mice that have white skin instead of wild-type mice with black skin (Fig. 4b). Remarkably, the efficiency of base editing in the mouse model by NG-ABEmax-KR was drastically higher than the original NG-ABEmax (Fig. 4c). While the original NG-ABEmax produced only 1 (9%) white and 2 (18%) mosaic (white/black) mice out of 11 in total, the NG-ABEmax-KR produced 15 (75%) white and 2 (10%) mosaic mice out of 20 in total. Next-generation sequencing of the target site from each mouse showed that the editing efficiency was greatly boosted at A3 (Fig. 4c, Supplementary Fig. 11), consistent with the phenotype results. Sanger sequencing further confirmed the successful editing events at the target site (Fig. 4d). We also investigated the effects of NG-ABEmax or NG-ABEmax-KR on mouse embryogenesis. The results showed that intracytoplasmic injection NG-ABEmax or NG-ABEmax-KR did not affect mouse embryogenesis but boosted gene-editing efficiency (Fig. 4b, Supplementary Fig. 12, and Supplementary Table 3).

Next, we sought to evaluate the performance of NG-ABEmax-KR in introducing disease-correcting mutations. An effective therapeutic strategy that greatly alleviate the clinical symptoms of β-globin diseases, such as sickle-cell anemia and β thalassaemia is to recapitulate British-type HPFH and induce fetal hemoglobin

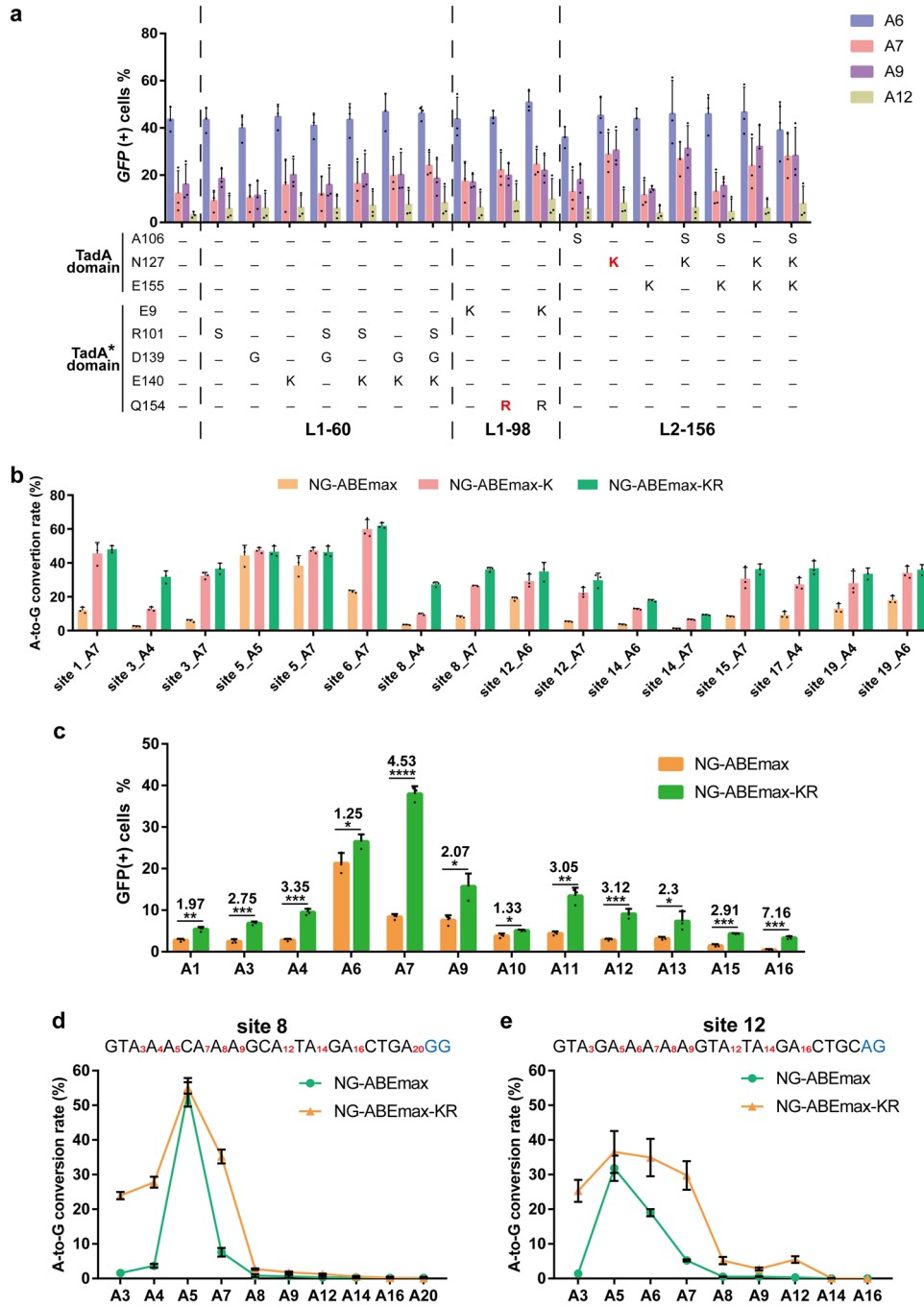

**Fig. 3 Engineering of the NG-ABEmax variants. a** Mapping the key residues for L1-60, L1-98 and L2-156 using *EGFP*-based reporter system. Editing efficiencies were quantified by flow cytometry. Error bars indicate mean ± s.d. (*n* = 3 independent experiments). **b** Boosted editing efficiencies of NG-ABEmax-KR at endogenous genomic sites. Error bars indicate mean ± s.d. (*n* = 3 independent experiments). **c** Summary of boosted editing efficiencies of NG-ABEmax-KR at different A base at the EGFP sites. Error bars indicate mean ± s.d. (*n* = 3 independent experiments). *$P < 0.05$, **$P < 0.01$, ***$P < 0.001$, ****$P < 0.0001$ by Student's unpaired two-sided *t* test. Exact *P* value of A1 = 0.002027 (A1 at 0.002027), A3 at 0.000347, A4 at 0.000167, A6 at 0.038443, A7 at 0.000012, A9 at 0.012894, A10 at 0.021788, A11 at 0.001507, A12 at 0.000910, A13 at 0.038361, A15 at 0.000149, A16 at 0.000345. **d, e** Increased editing window of NG-ABEmax-KR in site 8 and site 10. Each A base was highlighted in red. Error bars indicate mean ± s.d. (*n* = 3 independent experiments). Source data are available in the Source data file.

production in adults by promoter manipulation[32]. To this end, we designed a sgRNA to simultaneously mutate the *HBG1* or *HBG2* (*HBG1/HBG2*) promoters to activate their expression. We found that in HEK-293 cells, NG-ABEmax-KR efficiently installed the desired T•A to C•G mutations in *HBG1/HBG2* promoters, with a 4.32-fold increase in editing efficiency over NG-ABEmax (Fig. 4e).

**Off-target of NG-ABEmax-KR.** Enhanced gene-editing activity is generally associated with an elevated off-target activity. To ask whether NG-ABEmax-KR increases DNA off-target effects, we amplified fragments harboring the target site by PCR and performed sequencing to detect potential off-target editing. We observed a slight increase in editing at three off-target sites compared to NG-ABEmax (Supplementary Fig. 13). We also

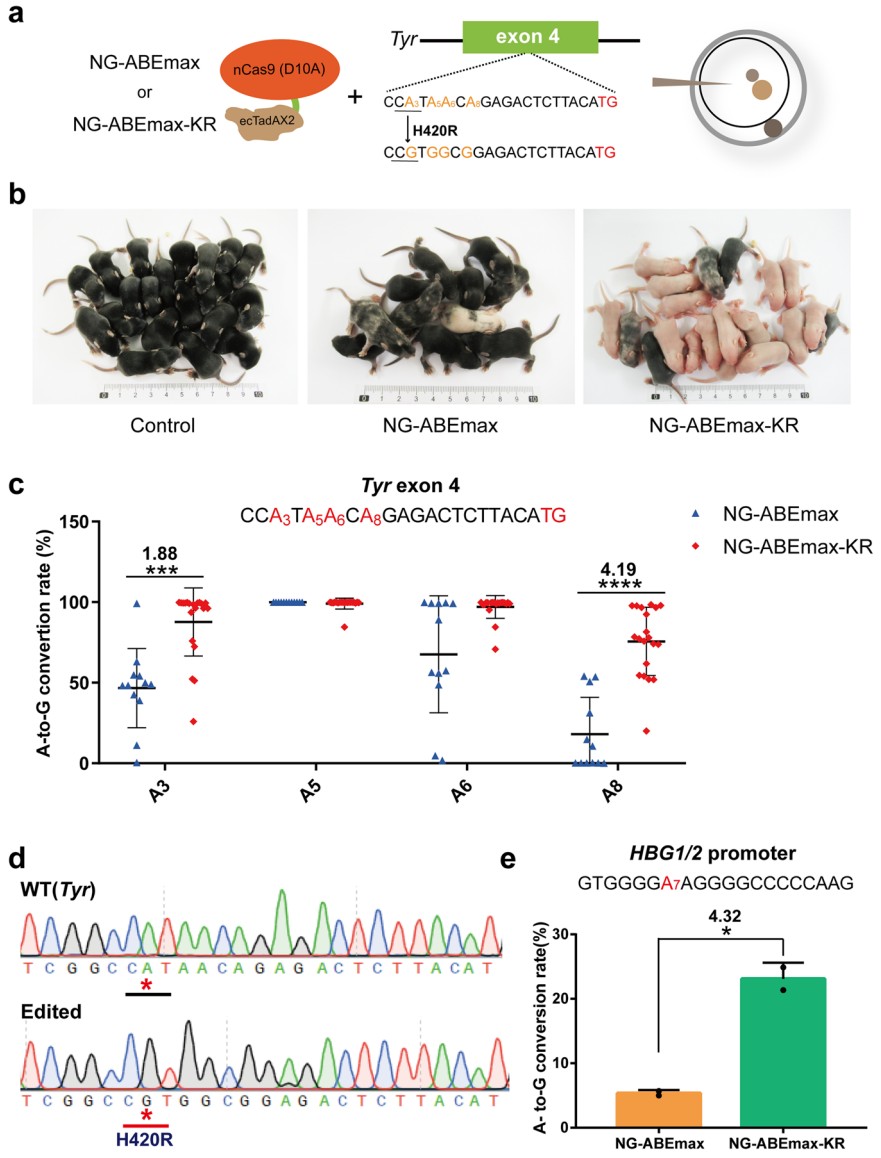

**Fig. 4 Application of NG-ABEmax-KR for the generation of mice disease models and human gene therapy. a** Schematic of editing activity comparison of NG-ABEmax and NG-ABEmax-KR in mouse embryos via zygote intracytoplasmic injection. **b** The newborn pups (days 10) produced by intracytoplasmic injection of NG-ABEmax or NG-ABEmax-KR mRNA and *Tyr* sgRNA. The *Tyr* mutant mice (H420R) are in white and the wild-type are in black, respectively. Control represents only ddH$_2$O injection. **c** Statistical analysis of on-target A-to-G base conversions induced by NG-ABEmax ($n = 12$), NG-ABEmaxKR ($n = 20$) in all pups. Data are mean ± s.d for the indicated numbers of mice. Each A base was highlighted in red. ***$P < 0.001$, ****$P < 0.0001$ by Student's unpaired two-sided $t$ test. Exact $P$ value of A3 = 0.00023, exact $P$ value of A8 = 0.00001. **d** Sanger sequencing chromatograms confirmed the editing events. The desired mutation was highlighted with red star. **e** Boosted editing efficiency (A7) of NG-ABEmax-KR at *HBG1/2*. Error bars indicate mean ± s.d. ($n = 2$ independent experiments). *$P < 0.001$ by Student's unpaired two-sided $t$ test. Exact $P$ value = 0.010120. Source data are available in the Source data file.

investigated the off-target effects of targeting *Tyr* gene for generation of mice disease models by the prediction of potential off-target sites, and observed no off-target at the majority of the predicted sites (Supplementary Fig. 14). We did observe off-target editing at certain other sites. These findings are consistent with the notion that NG-ABEmax-KR enhances the general catalytic activity of TadA that are manifest at both target and nontarget sites. To address whether NG-ABEmax-KR could increase the on-target:off-target editing ratios, we performed the analysis of on-target and its off-target. The results revealed a dramatic increase of NG-ABEmax-KR (up to 4037-fold at *HEK 3*-A6-OT4 of NG-ABEmax-KR VS 1432-fold of NG-ABEmax) at the off-target sites analyzed except one (*HEK 2*-A7-OT2, Supplementary Fig. 15). Together, these applications in mouse disease models and human

gene therapy show that NG-ABEmax-KR is an effective gene-editing tool that exhibits superior activity in both the human cells and animal models.

It is reported that ABE can also induce RNA off-target effects contributed by the wild-type TadA domain[20,21,33,34]. We investigated the RNA off-target editing of NG-ABEmax and NG-ABEmax-KR. According to the literature[20], we tested *CTNNB1* and *IP90*, two representative genes characterized with abundant mRNAs in HEK-293 cells. The results illustrated that RNA off-target editing of NG-ABEmax-KR is increased, compared with that of NG-ABEmax. While, plus two mutations (E59A in TadA, V106W in TadA*)[20], its off-target editing is similar as that of NG-ABEmax (Supplementary Fig. 16).

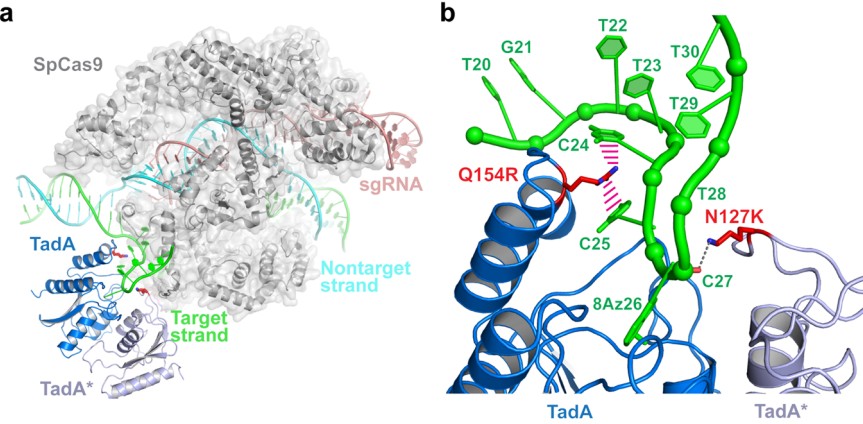

**Fig. 5 Molecular basis of enhanced editing efficiency of NG-ABEmax-KR. a** Overall structural model of NG-ABEmax-KR showing SpCas9 (gray), target-strand DNA (TS, green), nontarget-strand DNA (NTS, cyan), single-target RNA (sgRNA, red), TadA (blue) and TadA* (light blue). The side chains of N127K of TadA* and Q154R of TadA are shown as red sticks. **b** Modelled interactions between the key residues of NG-ABEmax-KR with TS DNA, colored as in (**a**).

**Molecular underpinnings of the enhanced editing efficiency of NG-ABEmax-KR.** To gain a mechanistic understanding of NG-ABEmax-KR's enhanced editing efficiency, we performed structural modeling. Structural modeling suggests that substitutions of the DNA-facing side chains of N127 of TadA* and Q154 of TadA with longer, positively charged side chains may create stabilizing interactions with the editing window, potentially augmenting catalysis (Fig. 5a, b). Specifically, an N127K substitution is expected to bring its ε-ammonium group within hydrogen-bonding distance (~3.6 Å) with the backbone nonbridging oxygen (OP1) of C27, immediately 3′ of the deamination site. This notion is consistent with our observation that N127R had a comparable impact on editing activity as N127K while N127H did not. On the opposite side of the DNA substrate, a Q154R substitution of TadA is projected to insert its guanidinium group between the nucleobases of C24 and C25, two nucleosides that immediately precede the deamination site (A26) on the 5′ side. Here, the Q154R guanidinium group would be located less than 3.5 Å away from both nucleobases of C24 and C25, and well positioned to make robust cation-π interactions to both, thereby stabilizing the DNA conformation to facilitate catalysis. Thus, in silico mutagenesis and structural modeling provide a straightforward explanation for the enhanced activity of NG-ABEmax-KR. Further, our findings provide a proof-of-principle that structure-informed rational redesign of the TadA-DNA interface, in particular, by introducing long, basic side chains at strategical locations, can effectively repurpose an enzyme interface originally evolved to recognize RNA hairpins, to recognize and accommodate topologically constrained, single-stranded DNA substrates.

## Discussion

ABE are particularly promising tools for the studies and applications of gene therapy, because two-thirds of human diseases are caused by single-nucleotide mutations, and about half of these can be corrected by A-to-G conversions[35]. However, ABE currently have limited applications due to low editing activity compared with CBEs. Due to the lack of a clear understanding of the artificial TadA-DNA interface within ABE, rational design-based activity optimizations have not borne significant fruit. In this study, we established and applied an *EGFP*-based screening system to efficiently evolve and select NG-ABEmax variants with desired traits, such as those with increased activity. Using this system, we obtained several mutants (NG-ABEmax-SGK, NG-ABEmax-R, and NG-ABEmax-K) that possess higher editing activity (Figs. 2–4). Besides higher editing efficiency, these three variants are able to effectively editing at certain inaccessible

positions by the wild-type NG-ABEmax (Fig. 2). Among the three variants, NG-ABEmax-K exhibited the best performance at most sites we examined (Fig. 3b, Supplementary Figs. 6, 7). To further enhance the activity, NG-ABEmax-KR was created by the combinatorial design using the identified driver mutations (Fig. 3b, c, Supplementary Fig. 9). Compared with NG-ABEmax, NG-ABEmax-KR showed superior editing activity and an expanded editing window (Figs. 3, 4).

The Cas9 element in ABE has been extensively studied as a limiting factor for PAM recognition[36–42]. However, some studies have illustrated that Cas9 elements are also responsible for the editing window of ABE, which indicates that the Cas9 element may also be selected for the evolution of ABE[36,37,43]. In addition, the sequence or structure of the sgRNA may also modulate the activity of ABE, suggesting it could be further optimized[10]. As for DNA and RNA off-target effects, we characterized the fidelity of NG-ABEmax and its variants (Supplementary Fig. 13, 14, 16). In addition to editing canonical base A by ABE, multiple studies have found that ABE can also edit base C at a low level[12,33,44,45]. Nonetheless, our study did not observe off-target editing products involving base C editing. The precise specificity of the NG-ABEmax variants awaits further comprehensive characterization.

Base G at the position 1 of sgRNA would promote RNA pol III transcription from the U6 promoter, while it is not clear whether G at position 1 of sgRNA would affect ABE efficiency or not. As some of 12 sgRNAs do not start with a G (Fig. 1d, e), we generated the plasmids with the same target sequence except the first nucleotide (A/C/T/G) and target A base in 3 (A3) or 4 (A4) with the corresponding sgRNA sequence. The plasmids and NG-ABEmax-KR were transfected into HEK-293 cells and then we used NGS to analyze the editing products. The results showed with NG-ABEmax-KR, at A3, the first nucleotide (A/C/G) triggered the robust editing; at A4, only C/G have similar performance (Supplementary Fig. 17). Our results indicated that to achieve the robust ABE mediated editing, C/G at position 1 of sgRNA may be highly suggested.

While our study was being prepared for publication, Richter and colleagues reported directed evolution of ABEmax based on phage-assisted non-continuous and continuous evolution (PANCE and PACE), which produced ABE8e[46]. Based on the synthetic library of TadA sequences that contains all 20 canonical amino acid substitutions at each position of TadA, Gaudelli and colleagues identified ABEmax variants (ABE8s) with higher activity[47]. Notably, we observed that the Q154R mutation that we selected was also identified in ABE8s but not ABE8e, suggesting that

directional evolution analyses based on different strategies may converge on the same critical sites. Our high-activity mutations do not overlap with those reported in the ABE8e[46,47]. A comparative analysis of the editing efficiencies of ABE variants showed that NG-ABEmax-KR possesses the similar activity, as that as ABE8s. Meanwhile, ABE8e has slightly higher activity than that of NG-ABEmax-KR and ABE8s (Supplementary Fig. 18a). We also compared the corresponding protein expression levels and nuclear localization (Supplementary Fig. 18b). The results of Western blotting revealed that, compared with ABEmax, all the mutants except ABE8.13-d have relative lower level of protein expression levels, especially ABE8e. We also performed immunostaining to detect the nuclear localization and the results indicated no notable difference of these mutants (Supplementary Fig. 18c). These results may yield additional insights into the path leading to improved efficiency and a combination of these mutations in the same enzyme may achieve synergistic effects as we observed in this study. Recently, the structure of ABE8e has been reported, which provides additional insights for the further rational design of ABE[48].

Taken together, in the present study, we created a high-throughput directional screening system to evolve, select and assess ABE activity in human cells, and identified NG-ABEmax variants with significantly increased editing activity. This system can be readily applied to evolve additional ABE traits besides enhanced activity, such as access to previously inaccessible editing sites, or modulation of off-target effects. Finally, our analysis provides a mechanistic rationale for the observed activity enhancement, which can guide further optimizations of the TadA-DNA interface.

## Methods

**Plasmid construction**. The original plasmid NG-ABEmax expressing the TadA-TadA*-nSpCas9-NG fusion protein was obtained from Addgene (Addgene plasmid #124163). The plasmid pSin-EGFP containing an *EGFP* gene, *IRES* and *Puromycin* genes were generated was generated from pSIN-EF2-Lin28-Puro(obtained from Addgene; #16580) using EcoR I and BamH I restriction enzyme sites[22]. The sgRNA expression cassettes were generated by standard protocol. Desired point mutations were introduced into the coding sequence of *EGFP* by PCR to generate *EGFP*-variant, *dEGFP1*, and *dEGFP2*. All plasmids were confirmed by Sanger sequencing. The oligonucleotide sequences used for plasmid construction in this study are listed in Supplementary Tables 4–9.

**Construction of NG-ABEmax mutant libraries**. NG-ABEmax libraries were generated using the protocols, which described previously[49]. Specifically, for the library 1, the NG-ABEmax plasmids were digested with BseRI, and the digested products (backbone) was subsequently purified (8277 bp). Then, GeneMorph II Random Mutagenesis Kit (Agilent) was used to perform error-prone PCR on the TadA* domain in the NG-ABEmax plasmid sequence, and the PCR products were purified as fragments (575 bp). The backbone and fragments were In-fusion assembled (Supplementary Fig 3a). For the library 2, the backbone plasmids were digested with SacII and BamHI (the final products of 8177 bp). Then, fragments (730 bp) of error-prone PCR on the TadA domain were obtained using the same method (Supplementary Fig 3a). The resulting library 1 and library 2 were transformed into electrocompetent *E. coli* DH10B and incubated on LB plates with Ampicillin (0.1 g/mL) at 37 °C overnight. A total of 1978 colonies (1428 of library 1 and 550 of library 2) were obtained. The plasmids from individual colonies were isolated. The oligonucleotide sequences used for libraries construction are listed in Supplementary Table 10.

**Cells and cell culture**. HEK-293 cells were obtained from ATCC (CAT#CRL-1573), and HEK293-PME cells containing MCS and EGFP expression cassettes were generated by lentiviral transduction[15]. HEK-293 cells were grown at 37 °C in 5% $CO_2$ in Dulbecco's modified Eagle's medium (Life Technologies, Carlsbad, CA) supplemented with 10% heat-inactivated fetal bovine serum, penicillin/streptomycin. HEK293-PME cells were cultured with additional puromycin.

**Transfection protocol, genomic DNA extraction, images, and flow cytometry analysis**. For NG-ABEmax variants screening experiments, HEK-293 cells were seeded at $0.9 \times 10^5$ cells per well on 24-well plates in DMEM medium. Twenty-four hours after seeding, cells were co-transfected with 200 ng NG-ABEmax variant plasmids, 100 ng sgRNA expression plasmids, 200 ng dEGFP1 plasmids, and 1.5 μl

TurboFect Transfection Reagent (Thermo Fisher Scientific). The medium was changed at 24 h post-transfection and flow cytometry analysis was performed at 48 h post-transfection. For experiments using HEK293-PME to compare the activity of NG-ABEmax variants, HEK293-PME cells were seeded at $0.7 \times 10^5$ cells per well on 24-well plates in the presence of puromycin. Twenty-four hours after seeding, cells were co-transfected with 200 ng NG-ABEmax variant, 100 ng sgRNA expression plasmids, and 1.5 μl TurboFect Transfection Reagent. Genomic DNA extraction of cells is subsequently performed using a genome extraction kit (Vazyme). Images were obtained at 24 and 48 h post-transfection, and cells were collected at 48 h post-transfection. Flow cytometry analysis was performed with FACSAria II (BD Biosciences). The raw data of screening has been summarized in Table S12 and S13.

**Comparison of the activity of NG-ABEmax variants using HEK293-PME cells**. The genomic DNA was used as A template for PCR amplification using corresponding primers and PCR products were subsequently purified and recovered (606 bp). Two hundred nanogram purified PCR products were treated with PstI (178 + 428 bp) or SalI (177 + 429 bp). Finally, the activity of NG-ABEmax variants was determined by agarose gel electrophoresis.

**Sanger sequencing for pJET colonies**. The sequence flanking the CRISPR target sites for *Try* was PCR amplified, and products were inserted into the vector pJET1.2 (CloneJET PCR Cloning Kit, Thermo Fisher Scientific). The ligated products were transformed into *E. coli*. The corresponding plasmids were isolated and sequenced on an ABI PRISM 3730 DNA Sequencer.

**Mice**. All animal procedures were carried out in accordance with the current guidelines of the Institutional Animal Care and Use Committee (IACUC) at the Center for Excellence in Molecular Cell Science, Shanghai Institute of Biochemistry and Cell Biology, Shanghai, China. B6D2F1 (C57BL/6 × DBA2) female mice were used as oocyte donors. The males (C57BL/6) were used to mate with B6D2F1 females to obtain zygotes. ICR females were used as pseudo-pregnant foster mothers. Mice were housed in individually ventilated cages (IVC) in an accredited specific pathogen-free facility under conditions with a 12 h dark-light cycle, a room temperature of 22 °C, 50% humidity, and free access to food and water.

**In vitro transcription**. The mRNA transcriptional templates of NG-ABEmax and NG-ABEmax-KR were amplified by PCR using Phanta Max Super-Fidelity DNA Polymerase (Vazyme), purified by the Universal DNA Purification Kit (TIANGEN) and then transcribed using the mMACHINE T7 ULTRA transcription kit (Invitrogen) following the manufacturer's instruction. The transcriptional template of *Tyr* sgRNA was amplified from Px330-mCherry plasmids[50](Addgene#98750) and transcribed in vitro using the MEGAshortscript T7 kit (Invitrogen) following the manufacturer's instructions. mRNA and sgRNA were subsequently purified with the MEGAclear Transcription Clean-Up Kit (Invitrogen), resuspended in RNase-free water, and then stored at −80 °C.

**Microinjection and embryo transfer**. The mixture of NG-ABEmax or NG-ABEmax-KR mRNA (100 ng/μl) and sgRNA (100 ng/μl) was diluted in RNase-free water, centrifuging at 4 °C, 13,400 g for 10 min and then injected into the cytoplasm of zygotes harvested from B6D2F1 females (mated with C57BL/6 males) using a micromanipulator (Olympus) and a FemtoJet microinjector (Eppendorf). The injected embryos were cultured in EmbryoMax KSOM Medium (Sigma-Aldrich) for 24 h to the two-cell embryos and 84 h to the blastocysts. Some two-cell embryos were transferred into oviducts of recipients at 0.5 days postcoitum (dpc). Recipient mothers delivered pups at 19.5 dpc and we analyzed phenotypes of offspring at day 10 after birth.

**Blastocyst genotyping**. Five mouse blastocysts were randomly picked from each group and directly lysed by 20 μl lysis buffer from the Mouse Direct PCR Kit (Bimake), incubated at 55 °C for 90 min and 95 °C for 5 min. Blastocyst lysate (about 2 μl) was used as a PCR template to amplify the editing site in *Tyr* using Phanta® Max Super-Fidelity DNA Polymerase (Vazyme). Products of PCR were purified by gel electrophoresis using Universal DNA Purification Kit (TIANGEN) and then performed Sanger sequencing and Next-generation sequencing (NGS).

**Measurement of protein expression levels**. To determine the expression levels of Cas9 protein, 800 ng plasmids expressing NG-ABEmax, NG-ABEmax-KR, NG-ABE8s(8.8d, 8.13d, 8.17d, 8.20d), and NG-ABE8e were transfected in $7.2 \times 10^5$ HEK-293 cells, respectively. Cells were collected 48 h after transfection to measure Cas9 expression levels. For Western blotting, anti-Cas9 antibody (ABclonal, Wuhan, China, #A14997) or anti-Actin antibody (ABclonal, Wuhan, China, #AC026) were used at a dilution of 1:10 000 or 1:50 000, respectively.

**Next-generation sequencing (NGS)**. Extracted genomic DNA from transfected cells or mice tails were performed by standard protocol. NGS library was constructed using genomic DNA as template through two rounds of PCR. First-step

PCR amplification of 100–220 bp sequences from on/off-target sites were performed using specific primers. For the second-step PCR amplification, we fixed the barcodes, index, and adaptor sequences to the first-step PCR amplification products. The second-step PCR amplification products were purified and pooled, and subsequently subjected to paired-end read sequencing using the Hiseq-PE150 strategy at Novogene (Nanjing, China). And the data collection of next-generation sequencing were used by Illumina MiSeq Control Software (v3.1). Finally, open-sourced "CRISPResso" software (version 1.0.10) was used to analyze the status of base editing. The oligonucleotide sequences used for NGS are listed in Supplementary Tables 11, 12.

**Molecular modeling**. Single amino acid substitutions are generated in silico using Coot[51]. Sidechain orientations are then geometrically optimized in Coot. Molecular graphics are prepared using MacPyMOL (Schrödinger, Inc).

**Statistics and reproducibility**. All data were expressed as mean ± SD Differences were determined by 2-tailed Student's $t$ test or Mann–Whitney test between two groups in Graphpad Prism 8.0.1 (244) . The criterion for statistical significance was *$P < 0.05$, **$P < 0.01$, ***$P < 0.001$, ****$P < 0.0001$, ns not significant ($P > 0.05$). Data shown is from on experiment (Figs. 1b and 2d; Supplementary Fig. 1b, 4, 12a, and 18b).

**Reporting Summary**. Further information on research design is available in the Nature Research Reporting Summary linked to this article.

## Data availability

The data that support the findings of this study are available in the Supplementary Materials. The deep sequencing data are available in the NCBI Sequence Read Archive (SRA) under accession code PRJNA714183 (SRA: SRR13958607- SRR13958611; SRR15440576-SRR15440577; SRR15440808-SRR15440809; sample accession number, SAMN 18291703); PRJNA713866 (SRA: SRR13945386-SRR13945395; SRR13948600-SRR13948607; SRR13950186-SRR13950197; SRR13950619-SRR13950626; SRR15441706; sample accession number, SAMN 18273806). Pre-processed data is available upon request. Source data are provided with this paper.

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

## Acknowledgements

We thank our group members for technical assistance. This work was supported by grants from National Key R&D Program of China [2018YFA0107304], National Natural Science Foundation of China (81201181), Science Technology project of Zhejiang Province (2017C37176), and Project of State Key Laboratory of Ophthalmology, Optometry and Visual Science, Wenzhou Medical University (J02-20190201), Shanghai Super Postdoctoral Incentive Program, China National Postdoctoral Program for Innovative Talents (BX20200348) and by the Intramural Research Program of the NIH, The National Institute of Diabetes and Digestive and Kidney Diseases (NIDDK) (ZIADK075136).

## Author contributions

F.G. and J.S.L. designed research, J.F., Q.L., X.L., T.T., X.L., J.N.L., and X.Y. performed research, J.F., Z.S., J.Q., J.Z., J.S.L., and F.G. performed data analyses, and J.Z., Q.L., and F.G. wrote the manuscript. All authors have read and approved the final manuscript.

## Competing interests

The authors declare no competing interests.
