## [Peer Review File · Nature Communications]

Reviewers' Comments:

Reviewer #1:

Remarks to the Author:

Review of

Human cells based directed evolution of adenine base editors with improved efficiency
Fu et al. 2021

Main comments

The authors undertook large-scale screening of ABE mutants in HEK293 cells revealing ABE variants with significantly enhanced activity and wider editing window over the originally reported ABE7.10max. The study is well designed, convincing and the new base editing tool could be of broad use to the scientific community, but any advantage over more recent ABE variants including ABE8s and ABE8e is not explored.

Unusually, the authors undertook screening by direct expression in mammalian cells. The authors should demonstrate that screening in mammalian cells offers advantages over screening for ABE mutants in bacteria (e.g. Gaudelli and Liu's original description of ABE, Nature, 2017), as bacterial screening is more scalable.

Perhaps screening in mammalian cells gives more insight into mammalian expression levels and editor protein localization than a bacterial system? This may explain why some of the enhanced activity mutants may differ from those described by Gaudelli et al and Richter et al Nature Biotech 2020.

I would suggest the authors compare protein expression levels, nuclear localisation and activity of their enhanced activity ABE mutant to those described by Gaudelli et al and Richter et al Nature Biotech 2020 (ABE8e and ABE8s).

The authors do not address RNA off-target editing of the new variants. This should be experimentally addressed, and compared to improved specificity variants such as the V106W mutant.

Minor comments

- Title could be Human cell based, rather than human cells based?
- I think the abstract claim that ABEs are not active enough for efficacious applications is too strong as they have been already widely used successfully in vitro and in vivo. Figure 1 shows > 50 % editing.
- Figure 1 – the authors claim that the EGFP variant fluorescent signal is not influenced by the silent and coding mutations introduced, but the intensity is not quantified. They should use flow cytometry to compare fluorescence intensity vs WT EGFP, for example in a histogram plot.
- Brookhouser BMC Biology 2020 use a similar stop reversion system for ABE, and Coelho et al BMC Biology 2018 have a BFP to GFP reporter for base editing. Authors should reference these.
- Line 83 – RNAase?
- Figure 1 – are these data from one biological experiment. Please note how many independent experiments in the legend.
- Figure 1f – it seems that guide A3 does not perform well. The primers show that some of the authors guides do not start with a G to promote RNA polIII transcription from the U6 promoter. Is this why A3 has not worked well, whereas A6 has (has a G at position 1). Authors should address this.
- Figure 2b – this is not a clear depiction of the screening data. The authors should at least make the raw data available in the supplement.
- "Nicole"? Perhaps the authors mean Nicole Gaudelli, so should write Gaudelli et al not just Nicole. Line 323.
- The authors claim that they have tested almost 2000 variants, but the library from error-prone PCR has only been validated by Sanger sequencing 40 clones, so the authors cannot be sure how many unique mutants have been screened without performing NGS on the libraries.

Reviewer #2:

Remarks to the Author:

To authors:

In the present manuscript titled "Human cells based directed evolution of adenine base editors with improved efficiency", the authors aimed to generate high-activity NG-ABEmax variants with a directional screening system in human cells. The authors first established a stop-codon reversion-based EGFP-reporter system and constructed two random variant libraries by error-prone PCR. Then, they combined these two systems to generate a series of high-efficiency NG-ABEmax variants. By combinatorial optimization of these variants the authors finally got the NG-ABEmax-KR variant showing superior editing activity and expanded editing window. To further assess the efficiency of NG-ABEmax-KR, the authors applied it in mouse disease models and human gene therapy and demonstrated its superior efficacy. This work established and applied an EGFP-based screening system to efficiently evolve and select NG-ABEmax variants with increased activity. Nevertheless, I would like to see the following points to be addressed in the manuscript before its publication in Nature Communication.

Comments/Suggestions

1. In this work, the author described "We found that 12 (8 of library 1 and 4 of library 2) out of the 1978 tested variants exhibited significantly elevated activity at A12, showing at least a 3-fold increase in editing activity (8.9%-11.25%), compared to the wild-149 type NG-ABEmax (2.62%-3.4%) (Fig. 2b)", but the photograph in Fig. 2b didn't show the mentioned editing efficiency results. Therefore, some sequencing data should be provided to show the comparison of editing activity at A12 site.
2. To establish a new NG-ABEmax variant, it was expected to comprehensively characterize its genome editing activities, including the editing efficiency, the editing window, the possible by-product, etc. .
3. Interestingly, the author successfully applied NG-ABEmax-KR to mouse disease models and human gene therapy. In general, the safety of base editor is always a key concern. Therefore, to assess the RNA off-target effects of NG-ABEmax-KR is extremely expected, although the author had analyzed the potential off-target editing in endogenous genetic sites
4. The author showed the results of newborn pups produced by intracytoplasmic injection of NG-ABEmax or NG-ABEmax-KR directly. In general, the editing efficiency of NG-ABEmax or NG-ABEmax-KR in mouse embryos should be described for understanding its possible effects on embryogenesis in Figure 4.

The point-by-point response

Reviewer #1

Main comments

Point 1. The authors undertook large-scale screening of ABE mutants in HEK293 cells revealing ABE variants with significantly enhanced activity and wider editing window over the originally reported ABE7.10max. The study is well designed, convincing and the new base editing tool could be of broad use to the scientific community, but any advantage over more recent ABE variants including ABE8s and ABE8e is not explored.

Author: We appreciate the reviewer's comments. Our directional screen platform based on human cells and identified mutants may be further utilized for the engineering of ABE (i.e., high-fidelity mutants, editing-window shifted editor or minimized by-stander mutation). An ongoing project for these purposes is being performed.

Point 2. Unusually, the authors undertook screening by direct expression in mammalian cells. The authors should demonstrate that screening in mammalian cells offers advantages over screening for ABE mutants in bacteria (e.g. Gaudelli and Liu's original description of ABE, Nature, 2017), as bacterial screening is more scalable. Perhaps screening in mammalian cells gives more insight into mammalian expression levels and editor protein localization than a bacterial system? This may explain why some of the enhanced activity mutants may differ from those described by Gaudelli et al and Richter et al Nature Biotech 2020.

I would suggest the authors compare protein expression levels, nuclear localisation and activity of their enhanced activity ABE mutant to those described by Gaudelli et al and Richter et al Nature Biotech 2020 (ABE8e and ABE8s).

Author: We acknowledge the reviewer's suggestion. Actually, we are more familiar with the human cells based screening system than that with the bacterial system. In 2014, we developed EGFP based reporter system (PMID: 24956376) and took advantage of it for the screening of high-fidelity SaCas9 and enhanced activity FnCas12a and new Cas12a member for genome editing (PMID: 32644995, PMID: 33567342 and PMID: 33715215). Recently, Hendel and Shoulders have summarized the directed evolution in mammalian cells and comprehensively illustrated its advantages and challenges (PMID: 33828274).

To explore the difference of our ABE mutants with ABE8e and ABE8s, we compared the activity, protein expression levels and nuclear localization of their enhanced activity ABE mutant to those described.

A comparative analysis of the editing efficiencies of ABE variants show that NG-ABEmax-KR possesses the similar activity as that as ABE8s. Meanwhile, ABE8e has slightly higher editing activity than that of NG-ABEmax-KR and ABE8s. We also compared the corresponding protein expression levels. The results of Western blotting

revealed that, compared with ABEmax, all the mutants except ABE8.13-d have a relatively lower level of protein expression levels, especially ABE8e. We also performed immunostaining to detect the nuclear localisation and the results indicated no notable difference of these mutants (Supplementary Fig. 16).

Supplementary Fig. 16 Activity, protein expression levels and nuclear localization of ABE variants. **a.** Base editing induced by ABEs at different A base in *EGFP*. Error bars indicate mean \pm s.d. ($n = 3$ independent experiments). **b.** Western blot analysis of ABE variants via the recognition with specific Cas9 antibody. **c.** The relative quantity of Cas9. β -Actin has been used as an internal control. **d.** Nuclear localization of ABE variants. HEK-293 cells were transfected with the plasmids, fixed 24 hours post-transfection and stained with the antibody against Cas9 (red). DAPI (blue) indicated the nucleus. Scale bar is 5 μ m.

Point 3. The authors do not address RNA off-target editing of the new variants. This should be experimentally addressed, and compared to improved specificity variants such as the V106W mutant.

Author: Thank you for this suggestion. To address this, we investigated the RNA off-target editing of NG-ABEmax, NG-ABEmax-KR and AW mutant in HEK-293 cells. According to the literature (PMID: 31086823), we tested *CTNNB1* and *IP90*, two representative genes characterized by abundant mRNAs in HEK-293 cells. We first transfected the HEK-293 cells with plasmids expressing for adenine base editors and then assessed the RNA off-target through NGS. The results showed that RNA off-target editing of NG-ABEmax-KR is increased, compared with that of NG-ABEmax. While RNA off-target editing of NG-ABEmax-KR plus two mutations (E59A in Tada, V106W in Tada*) is similar as that of NG-ABEmax (Supplementary Fig. 14).

Supplementary Fig. 14 RNA off-target editing of NG-ABEmax-KR and variants with two mutations (E59A in Tada, V106W in Tada*). **a.** The number of adenosines converted to inosine at a detectable level (>0.1%) of the corresponding mRNA. **b.** Average A-to-I RNA editing frequencies by NG-ABEmax, NG-ABEmax-KR and variants with AW (E59A in Tada, V106W in Tada*). Error bars indicate mean \pm s.d ($n = 3$ independent experiments).

Minor comments

Point 1. Title could be Human cell based, rather than human cells based?

Author: Thank you. We have corrected this in the revised manuscript.

Point 2. I think the abstract claim that ABEs are not active enough for efficacious applications is too strong as they have been already widely used successfully in vitro and in vivo. Figure 1 shows > 50 % editing.

Author: Thank you. We have modified this in the revised manuscript.

Point 3. Figure 1 – the authors claim that the EGFP variant fluorescent signal is not influenced by the silent and coding mutations introduced, but the intensity is not quantified. They should use flow cytometry to compare fluorescence intensity vs WT EGFP, for example in a histogram plot.

Author: Thanks for this suggestion. We provided the raw flow cytometry data. The fluorescence intensity data showed there is no notable difference between EGFP WT and EGFP variant (Fig. R1).

Fig. R1 Data of flow cytometry. The HEK-293 cells were transfected with the plasmids coding for EGFP (wild-type) and EGFP mutants. The flow cytometry analysis was performed to obtain the percent of EGFP positive cells.

Point 4. Brookhouser BMC Biology 2020 use a similar stop reversion system for ABE, and Coelho et al BMC Biology 2018 have a BFP to GFP reporter for base editing. Authors should reference these.

Author: Thank you. We have cited these papers in the revised manuscript.

Point 5. Line 83 – RNAase?

Author: Thank you. We have corrected it in the revised manuscript.

Point 6. Figure 1 – are these data from one biological experiment. Please note how many independent experiments in the legend.

Author: Thank you. We have pointed out how many independent experiments in the legend.

Point 7. Figure 1f – it seems that guide A3 does not perform well. The primers show that some of the authors guides do not start with a G to promote RNA polIII transcription from the U6 promoter. Is this why A3 has not worked well, whereas A6 has (has a G at position 1). Authors should address this.

Author: It is a nice point. While, to our knowledge, so far, it is not clear whether G at position 1 of sgRNA would affect ABE efficiency or not. As the reviewer pointed out that a G at the position 1 would promote RNA pol III transcription from the U6 promoter, the sequence itself may also affect the editing efficiency.

To address the reviewer's concern, we generated the plasmids with the same target sequence except the first nucleotide (A/C/T/G) and target A base in 3 (A3) or 4 (A4) with the corresponding sgRNA sequence. The plasmids and NG-ABEmax-KR were transfected into HEK-293 cells and then we used NGS to analyze the editing products. The results showed with NG-ABEmax-KR, at A3, the first nucleotide (A/C/G) triggered the robust editing; at A4, only C/G have similar performance (Fig. R2). Our results indicated that, to achieve the robust ABE mediated editing, G at position 1 of sgRNA would be highly suggested.

Fig. R2 Base editing efficiencies for sgRNA started with different first nucleotide.

a. The editing efficiencies of NG-ABEmax and NG-ABEmax-KR with four sgRNA targeted at position A3 in the *EGFP* (Figure 1d) in HEK-293 cells. **b.** The editing efficiencies at position A4 in HEK-293 cells. Error bars indicate mean ± s.d. (n = 3 independent experiments).

Point 8. Figure 2b – this is not a clear depiction of the screening data. The authors should at least make the raw data available in the supplement.

Author: Thanks for this suggestion. We have added the raw data of screening in the revised manuscript (Supplementary Table 12, 13).

Point 9. “Nicole”? Perhaps the authors mean Nicole Gaudelli, so should write Gaudelli et al not just Nicole. Line 323.

Author: Thank you. We have corrected it in the revised manuscript.

Point 10. The authors claim that they have tested almost 2000 variants, but the library from error-prone PCR has only been validated by Sanger sequencing 40 clones, so the authors cannot be sure how many unique mutants have been screened without performing NGS on the libraries.

Author: Thank you. We totally agree with the reviewer’s point. We have only estimated how many individual mutants are in the libraries. To get the accurate unique mutants number, NGS on the libraries should be performed.

Reviewer #2

Remarks to authors:

In the present manuscript titled “Human cells based directed evolution of adenine base editors with improved efficiency”, the authors aimed to generate high-activity NG-ABEmax variants with a directional screening system in human cells. The authors first established a stop-codon reversion-based EGFP-reporter system and constructed two random variant libraries by error-prone PCR. Then, they combined these two systems to generate a series of high-efficiency NG-ABEmax variants. By combinatorial optimization of these variants the authors finally got the NG-ABEmax-KR variant showing superior editing activity and expanded editing window. To further assess the efficiency of NG-ABEmax-KR, the authors applied it in mouse disease models and human gene therapy and demonstrated its superior efficacy. This work established and applied an EGFP-based screening system to efficiently evolve and select NG-ABEmax variants with increased activity. Nevertheless, I would like to see the following points to be addressed in the manuscript before its publication in Nature Communication.

Comments/Suggestions

Point 1. In this work, the author described “We found that 12 (8 of library 1 and 4 of library 2) out of the 1978 tested variants exhibited significantly elevated activity at A12, showing at least a 3-fold increase in editing activity (8.9%-11.25%), compared to the wild-type NG-ABEmax (2.62%-3.4%) (Fig. 2b)”, but the photograph in Fig. 2b didn't show the mentioned editing efficiency results. Therefore, some sequencing data should be provided to show the comparison of editing activity at A12 site.

Author: Thanks for this suggestion. We have added raw screening data in the revised manuscript (Supplementary Table 12, 13). And mutant's information is as shown below.

NG-ABEmax variants	Mut1	Mut2	Mut3	Mut4	Mut5	Mut6	Mut7	Mut8
L1-42	L144L (c.1083G>A)	Q153R						
L1-50	R100R (c.951C>G)	E133D						
L1-60	R100S	D138G	E139K					
L1-66	E2A	M60L	T78S	T82T (c.897A>G)	S96S (c.939T>A)	V101V (c.954G>A)	M125L	M150L
L1-79	N36S	R97R (c.942G>A)	T132S	Q153R				
L1-98	E8K	Q153R						
L1-240	E42V	W44R	I135I (c.1056C>T)	M150L	D166E			
L1-244	N126K							
L2-106	R149G							
L2-151	V32M	Q70Ter	N126K					
L2-156	A105S	N126K	E154K					
L2-159	Y80Y (c.297T>C)	Q158H						

Point 2. To establish a new NG-ABEmax variant, it was expected to comprehensively characterize its genome editing activities, including the editing efficiency, the editing window, the possible by-product, etc. .

Author: Thanks for this comment. We have addressed it in our revised manuscript (the editing efficiency in Fig.3b, c, Fig.4c, 4e and Supplementary Fig.8, 11b; the editing window in Fig. 3d, e and Supplementary Fig.9; the possible by-product in Supplementary Fig.12, 13, 14).

Point 3. Interestingly, the author successfully applied NG-ABEmax-KR to mouse disease models and human gene therapy. In general, the safety of base editor is always a key concern. Therefore, to assess the RNA off-target effects of NG-ABEmax-KR is extremely expected, although the author had analyzed the potential off-target editing in endogenous genetic sites

Author: Thank you for this point. To address this, we investigated the RNA off-target editing of NG-ABEmax, NG-ABEmax-KR and AW mutant in HEK-293 cells. According to the literature (PMID: 31086823), we tested *CTNNB1* and *IP90*, two representative genes characterized by abundant mRNAs in HEK-293 cells. We first transfected the HEK-293 cells with plasmids expressing for adenine base editors and then assessed the RNA off-target through NGS. The results showed that RNA off-target editing of NG-ABEmax-KR is increased, compared with that of NG-ABEmax. While RNA off-target editing of NG-ABEmax-KR plus two mutations (E59A in TadA, V106W in TadA*) is similar as that of NG-ABEmax (Supplementary Fig. 14).

Point 4. The author showed the results of newborn pups produced by intracytoplasmic injection of NG-ABEmax or NG-ABEmax-KR directly. In general, the editing efficiency of NG-ABEmax or NG-ABEmax-KR in mouse embryos should be described for understanding its possible effects on embryogenesis in Figure 4.

Author: We appreciate this suggestion. To understand the effects of NG-ABEmax or NG-ABEmax-KR on mouse embryogenesis with highly editing efficiency, we have added the results of developmental rate of pre-implantation embryos and newborn pups produced by directly injecting vehicle, NG-ABEmax or NG-ABEmax-KR into zygotes (Fig. 4b, Supplementary Fig. 11 and Supplementary Table. 11). The results showed that intracytoplasmic injection NG-ABEmax or NG-ABEmax-KR did not affect mouse embryogenesis but boosted gene editing.

Supplementary Fig.11 Editing efficiencies of NG-ABE_{max} or NG-ABE_{max}-KR in mouse embryos. **a**, Images of pre-implantation embryos obtained by injection of NG-ABE_{max} or NG-ABE_{max}-KR mRNA and sgRNA to target *Tyr* into zygotes. Control represents only ddH₂O injection. Scale bar, 100 μ m. **b**, Statistical analysis of on-target A-to-G base conversions induced by NG-ABE_{max} or NG-ABE_{max}KR in all blastocysts. Total embryos from each group were randomly divided into five groups; Data are mean \pm s.d. for five groups of blastocysts. Each A base was highlighted in red. *** $P < 0.001$ by Student's unpaired two-sided t-test.

Reviewers' Comments:

Reviewer #1:

Remarks to the Author:

The authors have addressed my major concerns on comparing to current ABE technologies in terms of efficiency, localisation and expression, and importantly, assessing RNA off-target editing.

I suggest the authors should incorporate ALL the figures included in the response to the reviewers comments in the manuscript figures themselves or supplement.

Please remove the graph showing quantification of the Western blot results as a bar chart for expression of base editor expression - this is meaningless without several repeats.

I also suggest that the results included after revision are not exclusively described in the discussion, but in the results section.

I would then find the manuscript acceptable for publishing in Nature Communications. I commend the authors on the contribution to the biotechnology field.

Reviewer #2:

Remarks to the Author:

In this revised manuscript, Fu et al. well addressed the points raised by the reviewers. The manuscript has been substantially improved, and is suitable for Nature Communications.

The authors have addressed my major concerns on comparing to current ABE technologies in terms of efficiency, localisation and expression, and importantly, assessing RNA off-target editing.

AUTHOR: Thank you!

I suggest the authors should incorporate ALL the figures included in the response to the reviewers comments in the manuscript figures themselves or supplement.

AUTHOR: Thank you for your comment. We have added all the figures into the revised manuscript.

Please remove the graph showing quantification of the Western blot results as a bar chart for expression of base editor expression - this is meaningless without several repeats.

AUTHOR: Thank you for your comment. We have removed this panel.

I also suggest that the results included after revision are not exclusively described in the discussion, but in the results section.

AUTHOR: Thank you for your comment. We have relocated the corresponding figures except two into the results section. Due to some data are very limited (i.e. the influence of first G for editing efficiency; the comparison of ABE variants), we only show it in the discussion. Additional studies would be required to address it.

I would then find the manuscript acceptable for publishing in Nature Communications. I commend the authors on the contribution to the biotechnology field.

AUTHOR: We appreciate these comments.